# Use of Heat-Shock and Edible Coating to Improve the Postharvest Preservation of Blueberries

**DOI:** 10.3390/foods12040789

**Published:** 2023-02-13

**Authors:** Chunyan Liu, Jie Ding, Peng Huang, Hongying Li, Yan Liu, Yuwei Zhang, Xinjie Hu, Shanggui Deng, Yaowen Liu, Wen Qin

**Affiliations:** 1College of Food Science, Sichuan Agricultural University, Yaan 625014, China; 2College of Food Science, Sichuan Tourism University, Chengdu 610100, China; 3College of Food and Pharmacy, Zhejiang Ocean University, Zhoushan 316000, China; 4Department of Quality Management and Inspection and Detection, Yibin University, Yibin 644000, China

**Keywords:** blueberry, heat-shock treatment, edible coating, postharvest quality

## Abstract

The quality of blueberry fruit is easily altered after harvest. We investigated the regulatory mechanism of heat-shock (postharvest treatment) and edible coating (preharvest treatment) on the post-harvest physiological quality of blueberry from the perspective of physiological, biochemical and organoleptic characteristics. In our research, the optimal TKL concentration and the appropriate range of heat-shock temperatures were first screened based on actual application results, and then a combination of heat-shock temperature and TKL coating with significant differences in preservation effects was selected to investigate the effects of different heat-shock temperatures and TKL60 composite coating on post-harvest quality and volatile compound concentration of blueberries under refrigerated conditions. Our results showed that TKL with 60 mg/L thymol can retard the development of the degree of membrane lipid peroxidation and effectively reduce the incidence of fruit decay and the severity of blueberries infected with major pathogens at 25 °C. Meanwhile, heat-shock treatments were effective in maintaining the quality of blueberries, with a certain advantage from 45 °C to 65 °C after 8 d of storage at ambient temperature, but these treated groups were slightly inferior to TKL60 groups for fresh-keeping effect. Remarkably, the combination of heat-shock treatment and edible coating application could extend the shelf life of blueberries by 7–14 d compared to the results obtained with coating alone under low temperature storage. Specifically, heat treatment at 45 °C for 60 min after TKL60 coating (HT2) retarded the decrease in the levels of ascorbic acid, total anthocyanin, total acid and soluble solids. Gas chromatography–mass spectrometry hierarchical clustering analysis showed that this treatment also improved the aroma of the fruit, which maintained a certain similarity with that of fresh blueberries after 14 d. Principal component analysis (PCA) of the results of the evaluations carried out using an electronic nose (E-nose) and electronic tongue (E-tongue) showed that blueberries of the HT2 treated group did not show a large placement change of the PC1 distribution area from that of the fresh and blank control group. Accordingly, the combination of coating with heat-shock treatment can effectively improve the post-harvest quality and aroma compound concentration of blueberries, showing good application potential in storage and preservation of fresh fruits such as blueberries.

## 1. Introduction

With the increasing income and consumption levels of consumers, blueberries have increased in popularity due to their health benefits, nutritional value, and taste [1,2]. However, blueberry harvesting season is always hot and rainy (in southwest China), so blueberries are very perishable after picking, making the fruit intolerant to storage and reducing its commercial value. Therefore, there is an urgent need to identify effective preservation methods that can extend the shelf life of blueberry.

Several preservation methods are commonly used to improve the shelf life of blueberry, including low temperature storage [3], controlled-atmosphere (CA) techniques [4], irradiation treatment [5], essential oil fumigation [6], heat-shock treatment [7], and edible film coating [8]. Low-temperature storage delays senescence and helps to preserve fruit quality [9]. However, this preservation method is the most basic and is often applied in combination with other preservation methods in practice. CA can reduce blueberry respiratory metabolism, slowing down senescence [10]. Tina, et al. [11] conclude that the optimal controlled atmosphere under long-term storage is 5% CO_2_, 5% O_2_, and 90% N_2_ for the maintenance of weight and nutritional quality of the blueberry fruit ‘Liberty’. However, the sudden change of atmosphere could elicit a physical abiotic stress response in the fruit, negatively affecting quality. Irradiation treatment significantly reduced total aerobic bacteria and yeast on the fruit surface [12]. Villagra et al. [13] showed that UV-C irradiation could be an interesting tool to improve antioxidant potential in highbush blueberries, which can negatively affect fruit quality for fresh consumption. The essential oils carvacrol, anethole, cinnamaldehyde, cinnamic acid, perillaldehyde, linalool, and p-cymene can inhibit blueberry decay, with carvacrol, anethole, and perillaldehyde simultaneously enhancing antioxidant activity of the fruit [14]. However, essential oil fumigation had significant negative effects on several sensory attributes such as sourness, astringency, juiciness, bitterness, and blueberry-like flavor [6]. As a barrier against water and oxygen exchange, mechanical damage, and pathogen infection, edible coatings are widely used to maintain quality and prolong the shelf life of horticultural products [15]. A pullulan coating can reduce the number of bacteria and mold in blueberry fruit, delay fruit ripening, and reduce weight loss [16]. Previous studies have found that a thymol/Konjac glucomannan (KGM)/low acyl gellan gum (LAG) (TKL) coating can reduce the secondary damage of postharvest treatment to fruit, effectively improve the edible quality of blueberry and prolong the storage period [17]. Heat-shock treatment refers to a physical preservation method for postharvest fruits and vegetables in a short period under non-lethal high temperatures [18]. Hot water at 42 °C reduces the decay of kiwifruit caused by *Botrytis cinerea* during postharvest storage and resulted in lower malondialdehyde content and higher total phenolic content in kiwifruits, regardless of whether it was used together with 5 g/L (*w*/*v*) potassium sorbate [19]. Hot air combined with nanomaterial packaging effectively inhibited respiratory intensity and delayed the senescence of bayberries [20]. Fan et al. found that hot water treatment at 22, 45, 50, or 60 °C increased the commodity rate of blueberries after four weeks of storage [7]. This is similar to the results of our pre-experiment. However, studies have shown that the combination of heat-shock and coating treatment is more effective in maintaining fruit quality [21,22].

The materials used in heat-shock (HT) treatment and TKL coating technology are simple and low cost. In addition, the combination of HT treatment and TKL coating technology can increase the release of effective components in the coating and inhibit or kill pathogenic microorganisms. HT treatment can also soften blueberry epidermal wax, plug fruit epidermal pores, and slow down respiration, extending the storage period of blueberry. To the best of our knowledge, there is no available scientific literature on the use of edible coatings in combination with heat treatment to maintain the quality and extend the shelf life of blueberries. Therefore, the aim of the present study was to evaluate the effect of a combined treatment using TKL edible coating and HT on freshness preservation, flavor compounds, and physiological–biochemical parameters of blueberry.

## 2. Materials and Methods

### 2.1. Materials

Rabbiteye blueberry (*Vaccinium* ashei ‘Baldwin-T-117’) samples were harvested from a commercial orchard in the northern suburb of Chengdu during the growing season. *Alternaria alternata*, *Penicillium* sp., and *Botrytis cinerea* were provided by the Key Laboratory Department of Agro-Products Processing and Storage, Sichuan Agricultural University (Sichuan, China). The materials used in the TKL composite coating were as follows: KGM (Henan Wanbang Industrial Co., Ltd., Shangqiu, China), LAG (Dancheng Caixin Sugar Co., Ltd., Dancheng, China), Calcium stearyl lactylate (Henan Hainasen Food Technology Co., Ltd., Xingyang, China), Thymol (Shanghai Yien Chemical Technology Co., Ltd., Shanghai, China), Polyethylene glycol 6000 (Chengdu Colon Chemical Co., Ltd., Chengdu, China), Beta-cyclodextrin (β-CD) (Mengzhou Huaxing Biochemical Co., Ltd., Mengzhou, China), and sucrose fatty acid esters (SE-15) (Liuzhou Aegefu Food Technology Co., Ltd., Liuzhou, China). All other chemicals and reagents are analytical grade (Chengdu Cologne Chemical Reagent Company, Chengdu, China).

### 2.2. TKL Coating

#### 2.2.1. Fabrication of TKL Coating

The following process was used to prepare the KGM and LAG-based film-forming edible biopolymer coatings. First, 0.025% KGM and 0.05% LAG were dissolved in 1 L water at 70 °C and then dispersed in a colloid mill for 10 min. Second, 0.08% calcium stearoyl lactate was dissolved in boiling water until clear and then added to the colloid solution.

The following process was used to prepare Thymol/β-CD slow-release microcapsules (TM). Polyethylene glycol 6000 aqueous solution (48.61 g/L) was mixed with SE-15 sucrose fatty acid ester aqueous solution (10 g/L) in equal volumes as the obtained first emulsion at 90 °C. β-CD was dissolved at a 9.72% concentration in 1 L of the first emulsion at 90 °C, then 50% thymol ethanolic was added at the rate of 2.86% by weight of β-CD. After 3 h stirring at 40 °C and 600 r/min, the solution was spray-dried with a Mini Büchi Spray Dryer QIMO-8000Y from China to obtain TM. The mean drug loading of the microcapsules was found to be 26.10 ± 0.49 mg/g. KGM and LAG-based edible coatings were combined with 0.22% thymol microcapsule to make TKL, of which the amount of thymol acting on the fruits was 60 mg/L.

Preparation of the TKL compound coating: Five KGM and LAG-based edible coatings were formulated to contain 0.08, 0.15, 0.22, 0.31, and 0.38% TM, which contained 20, 40, 60, 80, and 100 mg/L thymol, respectively.

#### 2.2.2. Application of TKL Coating

The TKL coating was evenly coated on the surface of ripe fruits (2.5 mL/g) using agricultural atomizing sprayer (Model 280, China Weifang Motong Machinery Manufacturing Co., Ltd., Weifang, China). After the coating solution had dried completely, the blueberry fruit were picked.

### 2.3. Antifungal Activity and Antioxidant Activity of TKL Composite Coating

The antifungal and antioxidant properties of TKL solutions with different concentrations (20 mg/L-TKL20, 40 mg/L-TKL40, 60 mg/L-TKL60, 80 mg/L-TKL80, and 100 mg/L-TKL100) were evaluated. The experiment was repeated in triplicate.

#### 2.3.1. Antimicrobial Activity

The antimicrobial activity of TKL composite coating was evaluated using three strains (*Penicillium expansum*, *Alternaria nees*, and *Botrytis cinerea*). The growth-inhibitory activities of TKL were assayed by the hyphal radial growth rate of the fungi from mycelial plugs placed on Potato Dextrose Agar (PDA) containing different concentrations of TKL composite coating (20, 40, 40, 80, and 100 mg/L) at 25 °C. The following equation was used to calculate the relative mycelial growth inhibition rate:R (%) = 100 × [(A1 − 5) − (A2 − 5)]/(A1 − 5)(1)
where R is the relative mycelial growth inhibition rate, A1 is the colony diameter in the control group (mm), A2 is the colony diameter in the treatment group (mm), and 5 is the diameter of the mycelial cake (mm).

#### 2.3.2. Antioxidant Activity

The antioxidant activity of the composite membrane was studied using the DPPH radical scavenging method. A preparation of 0.66 mM DPPH methanol solution was placed in a 4 °C refrigerator for dark preservation. A 0.1 g sample was added to a 10 mL methanol solution and shaken for 3 h at 150 rpm/min at room temperature and the supernatant was retrieved for measurement. Thereafter, 0.5 mL of sample solution was added to 4.5 mL of DPPH methanol solution. After full oscillation, the solution was reacted in the dark for 30 min and the absorbance (As) of the solution was measured at 517 nm. The sample solution was replaced by methanol in the control group (Ac). Calculation of the DPPH free radical scavenging rate used the following formula:Free radical scavenging rate (%) = 100% × [1 − (As − As0)]/Ac(2)
where Ac is 4.5 mL DPPH solution mixed with 0.5 mL methanol solution absorbance value, As is 4.5 mL DPPH solution mixed with 0.5 mL sample solution absorbance value, and As0 is 4.5 mL methanol mixed with 0.5 mL sample solution absorbance value.

### 2.4. Effect of TKL Composite Coating on Postharvest Quality of Blueberry

#### 2.4.1. Blueberry Treated with TKL Composite Coating

Six treatment groups were prepared in this experiment (20 mg/L-TKL20, 40 mg/L-TKL40, 60 mg/L-TKL60, 80 mg/L-TKL80, and 100 mg/L-TKL100). The blueberry fruit without any treatment was used as the control group (CK). The second to sixth groups were the various TKL treatments. Three replicates were used in all treatment groups, with 10 boxes of fruit (125 g/box) per treatment.

We evaluated the decay rate, preservation effect, and degree of pericarp membrane lipid peroxidation in blueberries, 0 d, 4 d, and 8 d after treatment with TKL solution during simulated marketing at 25 °C. The sampling methods were conducted using a completely randomized design, with three replicates per treatment.

#### 2.4.2. Decay Rate

The decay rate of blueberry fruit was determined according to the extent of decay on the peel: 0 = no decay, 1 = 0–25% decay, 2 = 25–50% decay, 3 = 50–75%, and 4 = 75–100% decay. The decay rate was calculated according to the formula:Decay rate (%) = (Decay scale × proportion of fruits corresponding to each scale)/4*n* × 100(3)
where *n* is the total number of examined fruits in each sample. Three boxes filled with blueberry fruit were randomly selected for each treatment. Approximately 120 fruit specimens from each sample were counted.

#### 2.4.3. Malondialdehyde

The malondialdehyde (MDA) content in the peel was measured by means of the thiobarbituric acid assay using a UV-Vis spectrophotometer (UV BlueStar A, Beijing Leibotaike Instruments, Beijing, China) [23].

#### 2.4.4. Measurement of Cell Membrane Electrolyte Leakage

The cell membrane electrolyte leakage was determined using a conductivity meter (Model DDS-11a, Shanghai Scientific Instruments, Shanghai, China). Take 0.20 g peel (5 mm × 5 mm) into a conical flask and add 30 mL distilled water, placed in a vacuum dryer for 10 min to extract intercellular air. Slowly put into the air, water penetrates into the intercellular space, the leaves become transparent, and the intracellular solute is easy to exude. The conical flask was taken out and oscillated every few minutes. The conductivity L1 was measured after being kept at room temperature for 30 min. Then the plugged conical flask was transferred into boiling water for 20 min. The conductivity L2 was measured after cooling to room temperature. Each test sample was assayed in triplicate and the experiment was repeated three times. The cell membrane electrolyte leakage was calculated as follows:Cell membrane electrolyte leakage (%) = L1/L2 × 100

### 2.5. Effect of HT on Postharvest Quality of Blueberry

#### Blueberry Treated with HT

Six treatment groups were prepared in this experiment. The blueberry fruit without any treatment after picking was used as the control group (CK). The second to sixth groups were treated with 25 °C, 35 °C, 45 °C, 55 °C, 65 °C hot air for 60 min, respectively, (Electric blast drying oven, 101A-4, Changzhou Jintan Liang You Instrument Co., Ltd., Changzhou, China). Three replicates were used in all treatment groups, with 10 boxes of fruit (125 g/box) per treatment.

We evaluated the decay rate, MDA content, and REC in blueberries, 0 d, 4 d, and 8 d after treatment with HT during simulated marketing at 25 °C. The sampling methods were conducted using a completely randomized design, with three replicates per treatment.

### 2.6. Effect of a Combined Treatment Using TKL Coating and HT on Postharvest Quality of Blueberry

#### 2.6.1. Combined Treatment of TKL Coating and HT on Blueberry Fruit

The TKL coating was sprayed before blueberry picking. The experiment was divided into four groups: CK, HT1 (TKL coating + 25 °C), HT2 (TKL coating + 45 °C) and HT3 (TKL coating + 65 °C). After pre-cooling at 8 °C for 12 h, the control group (CK) without TKL was stored at 90% relative humidity and 2 ± 0.5 °C for 35 d. In the HT group, blueberry fruits coated with TKL were heated for 60 min (25, 45 and 65 °C, respectively) in an electric blast dryer oven (101A-4, Changzhou Jintan Liang You Instrument Co., Ltd., Changzhou, China) after harvest and samples were stored in the same environment as the CK group for 35 d. HT1 served as a positive control, as 25 °C was nearly the ambient temperature on the day of harvest. The freshness, physiological and biochemical characteristics, and flavor compounds of blueberries were monitored at 0 d, 7 d, 14 d, 21 d, 28 d, and 35 d using the methods described below. The edible quality studies were divided into three terms: early (0 d), mid (14 d), and late (28 d). Decay rate ≥10% or weight loss ≥1.5% was set as the criterion to end the commercial storage period [24].

#### 2.6.2. Effects of HT and TKL Coating Treatment on Freshness of Blueberry

(1)Weight loss

Weight loss was measured using a gravimetric method. The weight loss was calculated according to the formula:Weight loss (%) = (W0 − W1)/W0 × 100(4)

W0 is weight of blueberry before heat-shock treatment, and W1 is weight of blueberry after storage.

(2)Total count of yeast and mold

Fungi counts on the blueberry surfaces were performed according to ISO 7954 using the spread method [25].

#### 2.6.3. Effects of HT and TKL Coating Treatment on Physiological and Biochemical Characteristics of Blueberry

(1)Total acid

Sample juice was prepared with 50 g of blueberries using a high-speed tissue-smash machine (FK-A, Changzhou Jintan Huanyu Scientific Instrument Factory, Changzhou, China), and filtered through fast filter paper. The juice was analyzed for total acid (TA), and soluble solid content (SSC).

TA was determined using the AOAC 937.05 method and expressed as grams of citric acid per 100 mL.

(2)SSC

The proportion of SSC was determined using a hand refractometer (0–30%, Jining Anyuan Mechanical Equipment Co., Ltd., Jining, China), and the results were expressed in Brix (1 g sucrose in 100 g solution).

(3)Ethylene release rate

Ethylene levels were measured as described by Shi et al. [26]. Thirty fruits from each sample were placed in separate 250 mL jars for 1 h at room temperature. One mL air samples were then taken from each jar using a syringe and injected into a gas chromatograph (clarus 680, Perkin Elmer, Waltham, MA, USA) to measure the ethylene. The experiment is repeated three times. The results were represented in μL/kg·h terms.

(4)Respiratory intensity

The respiratory intensity (RI) was measured using the static method. Initially, 100 g of blueberries was placed in a 30 mm vacuum dryer with 20 mL 0.4 mol/L NaOH solution. After standing for 1 h, the NaOH absorbed the CO_2_ released by blueberry respiration. The NaOH solution was then removed from the vacuum dryer, and 5 mL saturated BaCl_2_ solution was added to 2 drops of phenolphthalein. This solution was titrated with 0.1 mol/L oxalic acid. Blank titrations were performed using the same method. The RI (CO_2_ mg/kg·h) was calculated according to the following formula:RI = (Volume of oxalic acid consumed by sample − volume of oxalic acid consumed by blank) × molar concentration of H_2_C_2_CO_4_ × 44/(sample weight × standing time)(5)

(5)*L*-ascorbic acid

The *L*-ascorbic acid (*L*-Vc) of blueberry fruits was determined using HPLC (U3000, Thermo Fisher Scientific, Waltham, MA, USA). The experimental method corresponded to a previously published protocol with minor modifications [27]. A 2 g sample was drawn with 50 mL of 20 g/L of metaphosphate solution by ultrasound for 5 min and centrifuged at 8000× *g* for 20 min at 4 °C. The supernatants were filtered through a 0.45 μm syringe filter, and 2.5 μL of the sample was injected into the HPLC system. The flow rate was 0.6 mL/min with a mobile phase consisting of 1% acetonitrile and 99% 0.01 M sodium potassium dihydrogen phosphate (adjusted to pH = 2 with orthophosphoric acid), in a Thermo Hypersil C18 5 μm 25 × 0.46 column. The eluents were monitored using a UV detector at 245 nm wavelength. Three biological replicates with three technical replicates for each were performed for each sample. The *L*-Vc was calculated by comparison with the values obtained from a standard curve.

(6)Anthocyanin content

Anthocyanin content was determined for 10 g of fruit samples by HPLC, in which delphinidin, cyanidin, petunidin chloride, pelargonidin, peonidin, and malvidin were used as universal samples. The sample extractant was prepared by mixing anhydrous ethanol, water, and high-grade concentrated hydrochloric acid in a volume ratio of 2:1:1. Anthocyanins in the sample were collected to a volume of 50 mL by ultrasound for 30 min. The samples were then hydrolyzed in a boiling water bath for 1 h. After cooling to room temperature, their volumes were brought back up to 50 mL. The supernatant was collected after centrifugation at 10,000 rpm for 10 min at 4 °C in a cryogenic centrifuge. The cryogenic centrifuge of the extracts was filtered through 0.45 μm filter membranes for HPLC analysis. Mobile phase: 1% formic acid–water (A) and 1% formic acid–acetonitrile (B); gradient elution, and gradient elution program of the mobile phase are presented in Appendix A, Table A1. The reduction of anthocyanins in each sample was separated on a C18 reverse-phase chromatography column at 48 °C and measured using DVD (wavelength = 530 nm). The anthocyanin content was calculated by comparing the values obtained from a standard curve. Because pelargonium pigment was not detected in the detection process, the anthocyanin content in the sample was the sum of the other five compounds.

(7)Texture profile analysis

Texture profile analysis (TPA) was performed using a texture analyzer (TMS-PRO, FTC, USA) equipped with a 250 N load cell. A cylindrical 4.5 mm diameter ebonite probe was used for the TPA test, which included hardness, springiness, and chewiness. The TPA setting was maintained at a test speed of 1 mm/s, with a trigger force of 0.375 N and deformation rate of 30%. Each treatment selected 50 fruits for measured.

(8)Analysis of color

The blueberry samples from each treatment were subjected to color measurements using a full-automatic color difference meter (CR-410, Shanghai Liang Yan Intelligent Technology Co., Ltd., Shanghai, China). The specified parameters were *L**, *a** and *b**. Each measurement was repeated seven times.

#### 2.6.4. Effects of HT and TKL Coating Treatment on Flavor Compounds of Blueberry

(1)Analysis of aroma components

Following previous experimental methods [28], blueberry aroma was determined from 2 g of blueberry juice samples of each group using GC-MS with a PerkinElmer Elite-5MS column (30 m × 0.25 mm × 0.25 µm; Perkin Elmer, Waltham, MA, USA). The temperature of the samples was maintained at 45 ± 1 °C during headspace collection. The temperature program for the GC run was as follows: 40 °C initial column temperature, increased by 5 °C/min to 100 °C, held for 1 min, increased by 10 °C/min to 140 °C, held for 2 min, increased by 10 °C/min to 250 °C, held for 5 min. Helium (99.9999%) was introduced at a flow rate of 1 mL/min with a split ratio of 5:1. The mass range was set to 35–400 *m*/*z*. Volatile compounds were identified by comparing the mass spectra found in the NIST library. The peak area normalization method was used to determine the relative content of the volatile components.

E-nose and E-tongue (FOX 4000 Alpha MOS, Toulouse, France) analysis was conducted. Blueberry fruits (20.0 g per group) were homogenized in 80 mL of 80% deionized water. Each treatment was repeated seven times. Samples of 2 g of blueberry juice were collected in headspace vials at 60 °C for 5 min, and then evaluated by electronic nose. The E-nose measurement parameters were as follows: sensor cleaning time of 180 s, injection time of 3 s, gas flow rate of 150 mL/min, and data collection time of 120 s. Blueberry juice samples (20 mL) were diluted 5-fold with 80 mL of deionized water and used to measure fruit taste using an E- tongue autosampler tray for the measurement sequence.

(2)Sensory evaluation

Sensory evaluation followed a previous method [29] where blueberries were subjected to sensory evaluation by 10 trained panelists (five male and five female). This value can be measured by a fuzzy synthetic evaluation model with the formula:W = Σ (j × gi)(6)
where j is the specific score of each grade index, and gi is the calculation result of matrix. Scores for smell, taste, tissue state, and fruit color score criteria were as follows: 9, excellent; 7, good; 5, normal; 3, poor; and 1, unacceptable. The sensory scoring criteria are shown in Table 1.

### 2.7. Statistical Analysis

In the post-harvest preservation experiment of blueberry, each treatment was sampled using three boxes (125 g/box), and each index was measured three times. The results were reported as mean ± standard deviation. Excel 2019 was used to calculate data statistics. Data were analyzed using SPSS.25. Origin Pro 2021 was used for plotting the graphs presented in the figures. One-way analysis of variance (ANOVA) and Duncan’s test were performed to detect possible differences among the different treatments, and significant differences were expressed as *p* < 0.05.

## 3. Results and Discussion

### 3.1. Antifungal Activity and Antioxidant Activity of TKL Composite Coating

It is reported that *Penicillium* [30], *Alternaria nees* [31], and *Botrytis cinerea* [30] are common pathogens of postharvest decay of blueberry. As shown in Table 2, TKL compound coating had inhibitory effects on *Penicillium expansum*, *Alternaria alternata*, and *Botrytis cinerea*, and the inhibitory effect was positively correlated with the concentration of thymol microcapsules, but the inhibitory effects of 60 mg/L, 80 mg/L, and 100 mg/L TKL compound coating showed no significant difference. Ding et al. [24] also reported that thymol had adverse effects on the growth of the *Penicillium expansum*, *Alternaria alternata*, and *Botrytis cinerea*. With the increase of thymol microcapsule addition, the antioxidant capacity of TKL composite coating increased, but there was no significant difference in the antioxidant capacity of 60 mg/L, 80 mg/L, and 100 mg/L TKL compound coating. In summary, 60 mg/L-100 mg/L TKL composite membrane liquid had excellent antifungal activities against pathogenic fungus and antioxidant properties.

### 3.2. Analysis of the TKL Composite Coating on Preservation of Blueberry

TKL composite membrane fluids could reduce the decay rate of blueberry during storage at room temperature (Figure 1). The decay rate of the control group was more than 10% on the fourth day of storage; there was no significant difference in decay rate between TKL20 and TKL40 groups and CK group. On the eighth day of storage, the fruits of all groups showed different degrees of decay, and the TKL60 group had the least decayed fruit and the lowest decay rate (10.33%). Interestingly, there was no significant difference in fruit decay rate between TKL100 and TKL20 at the end of storage. This may be explained by the fact that thymol is a natural compound exhibiting phytotoxic activity [32]. TKL60 group had the lowest decay rate during the whole storage period, indicating that 60 mg/L thymol could effectively inhibit fruit decay.

### 3.3. Analysis of the TKL Composite Coating on the Degree of Pericarp Membrane Lipid Peroxidation of Blueberry

The MDA content and cell membrane electrolyte leakage of CK group increased by 320.56% and 125.12%, respectively, throughout the storage period (Figure 2). The MDA content and cell membrane electrolyte leakage of the TKL composite coating group were lower than those of the CK group, indicating that thymol could inhibit the degree of membrane lipid peroxidation of the fruit. Research by Ding shows thymol alone can inhibit peroxidation of the membrane lipids present in blueberry peel after pathogen infection [24]. The MDA content and cell membrane electrolyte leakage of blueberry fruit in the TKL60 group were significantly lower than those in other treatment groups throughout the storage period, which indicated that TKL60 treatment inhibited the membrane lipid peroxidation of blueberry fruit and helped maintain the structural integrity of the peel cells. Compared with the control group, the membrane lipid peroxidation level of blueberry treated with TKL100 increased continuously in the middle and late storage. However, it should be noted that no significant differences in the MDA content were observed between any of the samples (that is the control and all TKL treatment groups), thereby indicating that thymol disrupts the cell membrane structure and functionality at high concentrations.

In summary, higher thymol concentrations are not necessarily better for the storage quality of blueberry. TKL60 composite coating had good antifungal and antioxidant properties and could inhibit the decay of blueberry fruit and the degree of membrane lipid peroxidation of peel. Therefore, TKL60 was selected for subsequent research.

### 3.4. Effect of HT Treatment on Blueberry Quality

The decay rate of blueberry decreased with the increase of heat treatment temperature (Table 3). At the end of storage, the decay rates of 25 °C, 35 °C, 45 °C, 55 °C, 65 °C groups were 6.66%, 24.39%, 30.83%, 41.27% and 44.96% lower than those of the control group, respectively. Fan et al. also compared the effects of heat treatment at different temperatures on the quality of highbush blueberry fruit. The results showed that the higher the temperature, the lower the fruit decay rate, which is similar to the results of this paper [7]. In this study, we also found that heat-shock treatment could reduce the degree of membrane lipid peroxidation of blueberry fruit, mainly manifested as the decrease of MDA content and cell membrane electrolyte leakage during blueberry storage. Similarly, the higher the heat-shock temperature, the lower the MDA content and cell membrane electrolyte leakage of blueberry fruit. However, there was no significant difference between 35 °C and 45 °C groups, 55 °C and 65 °C groups. A large number of studies have confirmed that the combination of heat-shock and coating treatment has a better quality on fruits. Hot air combined with chitosan coating treatment can improve total phytochemicals and antioxidant activities of Sanhua plum [22]. Research by Xue et al. shows combination of heat treatment at 50 °C and 1% CM-C solution was finalized as the best way to keep the fruits fresh [21]. Therefore, in order to study the effect of heat-shock combined with TKL coating treatment on the storage quality of blueberry, we selected 25 °C, 45 °C and 65 °C groups with the most significant difference and TKL combined treatment.

### 3.5. Effects of HT and TKL Coating Treatment on Freshness of Blueberry

#### 3.5.1. Weight Loss

Heat-shock treatment will cause the weight loss of blueberry (storage 0 d), and it is positively correlated with the heat-shock temperature. However, with the extension of storage time, the CK group without any treatment had the most loss of fruit quality, followed by the HT1 group, HT3 group, and finally HT2 group (Figure 3a). At the 35th day, the weight loss of the HT1, HT2, and HT3 group was 26.72%, 48.10%, and 37.08% of the control group (*p* < 0.05), respectively. The dehydration and wilting of blueberry during storage was caused by the respiration and transpiration of fruits after harvesting, while heat-shock treatment and coating could reduce the respiration and transpiration of fruits, thereby inhibiting the reduction of fruit weight loss rate [22,33]. In summary, HT2 treatment was the most beneficial for the maintenance of fruit weight. It has been reported that a combination of chitosan coating and heat-shock treatments could significantly reduce the weight loss of Akebia trifoliate fruit during cold storage [34].

#### 3.5.2. Decay Rate

Decay is a major cause of postharvest quality loss in fresh fruits [7]. As exhibited in Figure 1b, the decay rate of blueberry fruits gradually increased, while the control group had the highest decay rate during storage. Decay rate ≥10% was set as the criterion to end the commercial storage period. The decay rate of the control group was more than 10% at 14 d, while the decay rates of HT1, HT2, and HT3 groups were more than 10% at 21 d, 35 d, and 28 d, respectively. At the end of storage, moldy fruits appeared in all groups. The fruits of CK group were almost all moldy (Figure 3b), and the decay rate was 25.15%, while the decay rate of HT1, HT2, and HT3 groups were 21.5%, 10.55%, and 13.81%, respectively, which were lower than those of the control group (*p* < 0.05). The results showed that, compared with HT1 and HT3, the HT2 group showed better inhibitory effects and the storage period of 21 d could be extended. This phenomenon is related to the heat-shock temperature. Higher heat-shock treatment temperatures cause damage to the fruit while eliminating the microorganisms attached to the fruit surface, thus exacerbating the fruit decay [33]. This was consistent with the results of a previous report which found that the decay rate of ‘Hachiya’ persimmon fruit treated at 50 °C was lower than that at 55 °C [35].

#### 3.5.3. Total Count of Yeast and Mold

The total count of yeast and mold of blueberry fruit gradually increased during the storage period (Figure 3c). However, compared with the control group, the total count of yeast and mold of blueberry fruit after film coating treatment were significantly reduced (*p* < 0.05). Interestingly, at 0–14 d storage, the total count of yeast and mold of blueberry after different treatments showed CK > HT1 > HT2 > HT3. This may be due to microencapsulated essential oils that exhibited temperature-sensitive release. After the thermal treatment, the small degree of softening of the blueberry waxy bloom improved the penetration of the TKL coating into the fruit tissue, allowing enhanced access of thymol into the hydrophobic outer skin wax. Therefore, the complex retained significant antifungal action during low-temperature storage after thermal treatment at 45 °C or 65 °C for 60 min. The same results were obtained when including complexes of eucalyptus essential oil with β-cyclodextrin [36] and microencapsulation of garlic oil by β-cyclodextrin [37]. However, at 21–35 d storage, the total count of yeast and mold of blueberry after different treatments showed CK > HT1 > HT3 > HT2. The total count of yeast and mold in HT2 at the end of storage was 6.12 ± 0.03 lgCFU/g, which was significantly lower than that of the CK group. According to a previous study, the use of non-forced hot air to control *Botrytis cinerea* and *Penicillium expansum* infection proved effective, requiring 38–46 °C treatment for 12–96 h [38]. The combined application of heat-shock and edible coating resulted in a higher inhibition of postharvest pathogenic fungal growth when compared with non-thermally treated samples.

### 3.6. Effects of HT and TKL Coating Treatment on Physiological and Biochemical Characteristics of Blueberry

#### 3.6.1. Ethylene Release Rate

The ethylene release rate, as an important physiological index, can directly reflect the ripening and senescence of fruits and vegetables during storage. The results showed that the ethylene release rate of the four groups had an initial increasing trend followed by a gradual decline (Figure 4a). The peak of ethylene release in the CK group and HT1 group appeared on the 14th day of storage, which were 5.33 ± 0.12 µL/g h and 4.34 ± 0.45 µL/g ·h. The ethylene release rate of fruits in HT2 and HT3 groups was 3.57 ± 0.10 µL/g·h and 3.76 ± 0.07 µL/g·h, respectively, after 14 days of storage. The maximum ethylene release of fruits in HT2 and HT3 groups did not appear until 21 days of storage. During the whole storage period, the ethylene release was lower in HT group fruit than in the non-treated fruits, and the effect of the HT2 treatment group was the best. Therefore, it showed that appropriate temperature combined with coating treatment effectively delayed the senescence process of blueberry fruit.

#### 3.6.2. Respiratory Intensity

The respiration intensity of blueberry fruit in each group initially increased and then decreased, which was similar to the observed trend of ethylene release rate (Figure 4b). During 0–14 d of storage, the respiration intensity of fruits in CK and HT1 increased sharply, reached a peak at 14 d with peak values of 12.58 ± 0.20 mg CO_2_/(g·h) and 12.03 ± 0.30 mg CO_2_/(g·h), respectively, and then decreased. This may be related to the senescence of the fruit [39]. The respiratory intensity of HT2 and HT3 groups reached the peak at 21 d, and the peak values were 11.00 ± 0.32 mg CO_2_/(g·h) and 11.15 ± 0.44 mg CO_2_/(g·h), respectively. At the end of storage, the respiration intensity of fruits (HT1, HT2, and HT3 groups) after coating treatment was lower than that of CK group. This is likely because the TKL edible coating forms a molecular film barrier, reducing the availability of oxygen and the water vapor permeability, increasing the near-surface carbon dioxide, and reducing respiratory intensity. This conclusion is similar to that of previous studies [8,40]. HT2 and HT3 are the most effective ways to reduce respiratory intensity. Decreasing the respiration intensity of fruit can prolong its shelf life [41]. This shows that 45 °C and 65 °C heat-shock treatment can effectively delay the aging process of harvested fruit.

#### 3.6.3. Total Acid

TA decreased gradually with the extension of storage time (Figure 4c). This could be associated with increased consumption of organic acids during respiration with the extension of storage time [42]. Heat-shock and coating treatment effectively reduces the consumption rate of organic acids, thus delaying the decrease of total acid content. Our results indicated that the TA content of the CK group decreased the most rapidly, reaching a value of 3.62 ± 0.20 g/kg (60.65% reduction) after 35 d. For the HT1, HT2, and HT3 treatment groups, the corresponding rates were 56.07%, 48.08%, and 48.00%, respectively. It was also observed that 45 °C or 65 °C heat-shock treatment was the most effective in reducing the consumption rate of organic acids. Lv et al. compared the effects of heat-shock (HT), 1-methylcyclopropene (1-MCP), or their combination (HT + 1-MCP) on the quality of fresh jujube fruits during cold storage. The results showed that HT demonstrated the best preservation effect on jujube fruits, and better suppressed the decrease in TA content than other treatment groups [42].

#### 3.6.4. Soluble Solid Content

The trend of SSC content was similar to that of TA, and both show a downward trend (Figure 4d). Studies have shown that the decrease of soluble sugar and titratable acid in blueberry fruit will shorten the shelf life of fruit [43]. The SSC content of the fruit in the CK group decreased the fastest, decreasing by 47.89% at the end of storage. This may be due to respiratory consumption and macromolecular degradation. The SSC content of the fruit in the HT1 group decreased by 44.04%, which was significantly lower than that in the CK group (*p* < 0.05). This might be attributed to the fact that the edible film reduces water loss, effectively maintaining the TSS content inside the fruit [44]. The SSC content of fruits in HT2 and HT3 groups decreased by 37.75% and 41.79%, respectively, during storage, which showed that the combined treatment of heat-shock and coating film could more effectively inhibit the decrease of SSC content of blueberries. Heat-shock treatment will lead to the continuous conversion of starch and other polysaccharides into small soluble carbohydrates or some insoluble pectin transforming into soluble pectin [42]. Therefore, the decrease of SSC content slowed down after heat-shock treatment.

#### 3.6.5. *L*-ascorbic Acid

Both heat-shock and coating treatments significantly affected the respiratory intensity of fruits during storage, resulting in changes to sugar, acid, and other nutrients in blueberries. In the present study, the *L*-Vc content of blueberry showed a decreasing trend during storage (Figure 4e), which was consistent with the report of Ansorena et al. [45]. The *L*-Vc content of fruit in CK group was lower than that in other groups during the storage period. At 35 d storage, the *L*-Vc content of CK group was 5.34 ± 0.29 mg/100 g and the *L*-Vc content of HT1, HT2, and HT3 were 1.25, 1.43, and 1.41 times that of the CK group, respectively. This shows that 45 °C or 65 °C heat-shock combined with coating treatment delayed the decrease in *L*-Vc content during all storage periods and reduced the loss of nutrients after harvest.

#### 3.6.6. Anthocyanins

*L*-Vc and anthocyanins are important indices reflecting fruit quality. Anthocyanins in blueberries are gradually oxidized during storage and therefore exhibit a continuous decrease, similar to the trend observed for *L*-Vc (Figure 4f). Similarly, the anthocyanin content of CK group was lower than that of other treatment groups throughout the storage period. At the end of storage, the anthocyanin content of CK, HT1, HT2, and HT3 groups decreased by 35.82%, 26.63%, 20.81%, and 27.77%, respectively. The anthocyanin content of HT2 group (31.31 ± 0.04 mg/L) was significantly higher than that of other groups at 35 d of storage. Therefore, HT2 treatment can effectively protect anthocyanins and delay the decrease of anthocyanins. This may be related to the high TA content in HT2 treatment fruit. Studies have shown that the acidic environment was more conducive to the preservation of anthocyanins [46].

#### 3.6.7. Texture Profile Analysis

The TPA test results showed that firmness of CK and HT1 groups decreased from harvest to the end of storage (Figure 5). The fruit firmness of CK and HT1 decreased by 48.83% and 40.94%, respectively, during storage. Blueberry fruits subjected to hot air treatments at 45 °C or 65 °C, often softened more slowly than those stored at 25 °C or non-heated fruits (*p* < 0.05). After 14 d of storage, blueberries from HT2 increased slightly and were 24.93% firmer than those of the CK groups. The resilience of HT2 and HT3 was significantly higher than that of CK (*p* < 0.05) during storage. The results of Zhang et al. showed that 1-MCP and hot air treatments maintained the firmness and delaying softening of nectarine fruit [47]. The resilience and chewiness of fruits in each group decreased significantly during storage. The resilience of the CK group decreased from 1 ± 0.13 mm to 0.55 ± 0.02 mm during storage, and that of HT1 group decreased from 1.17 ± 0.07 mm to 0.59 ± 0.05 mm. Compared with the CK group, HT2 and HT3 had higher resilience values during storage (*p* < 0.05). The resilience values of HT2 and HT3 at 28 d were comparable to those of the CK group and HT1 group at 14 d (*p* > 0.05). Chewiness of fruits in CK and HT1 groups decreased the fastest and decreased by 70.33% and 65.89%, respectively after 28 d storage. In contrast, the chewiness of fruits treated with HT2 and HT3 decreased by only about 37% during the whole storage period, which maintains the chewiness of blueberry fruits more effectively. This may be because HT2 and HT3 maintains the high hardness of blueberry fruit. Xie et al. [48] shows that the firmness was strongly correlated with chewiness and resilience. These results demonstrate that the 45 °C or 65 °C hot-shock combined with coating film were successful in maintaining the texture properties of blueberry.

#### 3.6.8. Color

The results obtained from the skin color analysis are shown in Table 4. With the extension of storage time, the *L** value of blueberry fruit decreased, and the *a** and *b** values increased. In the CK group, *L** values decreased on average from 14.29 ± 2.15 to 9.05 ± 0.61, *a** values increased from 0.4 ± 0.15 to 1.46 ± 0.17, and *b** values increased from −2.54 ± 0.75 to −1.12 ± 0.24 units from 0 d to 28 d, and there were no significant differences between HT1 and CK. When heat-treated berries (HT2 and HT3) were compared with the controls, they exhibited significantly higher *L** and lower *b** values during the storage period (*p* < 0.05). In general, when compared with CK, the HT1, HT2, and HT3 blueberries exhibited delayed redness, as observed by the lower *a** values, which showed an increase over time in the control samples. During refrigeration, the red discoloration of blueberry flesh is considered an important reduction in quality and has been regarded by researchers as a chilling injury [3]. HT2 and HT3 treatments had a more pronounced effect on blueberry skin color compared with CK and were able to maintain fruit gloss (*p* < 0.05) and reduce the rate of fruit redness in mid to late storage, with HT2 showing the highest performance. This may be because the expression of a small heat-shock protein gene (VcHSP17.7) in blueberry fruit was induced by heat-shock treatment and gradually increased during subsequent low-temperature storage, which alleviated chilling injuries and enhanced the chilling tolerance of blueberry fruit [49]. This beneficial action of HSPs is possible because of their chaperone activity and functions as membrane stabilizers and ROS scavengers, or their ability to act synergistically with antioxidant systems [50].

### 3.7. Effects of HT and TKL Coating Treatment on Flavor Compounds of Blueberry

#### 3.7.1. Flavor

Figure 6a presents the fingerprint profile of taste, containing most of the electronic tongue feature information because the cumulative contribution rate of PC1 and PC2 was 89.198% > 85%. The samples stored for 28 d could be well separated on the two-dimensional PCA map, which indicated clear differences in taste between the different groups. All samples in the HT2 group and blank samples stored for 0 d and 14 d were in the same quadrant and close to each other in space. In particular, the position of HT2, stored for 28 d, almost overlapped with that of CK, stored for 14 d, indicating the highest similarity between these groups. The results revealed that the combination treatment of 45 °C heat-shock and TKL edible coating had no significant difference with respect to the taste profile of the blueberries during the entire storage period (*p* > 0.05), and the taste of the fruit was well preserved.

Figure 6b compares the breakdown of the odor fingerprint profile according to the variance contribution rates of PC1 (90.448% > 85%), as PC1 contained most of the electronic nose feature information. The PC1 of the electronic nose responses for the HT2 group exhibited slight variation throughout the storage period, with the PC1 being similar to the CK group stored for 14 d.

Figure 6c shows that volatile compounds of the blueberry fruits at different treatments, corresponding to 0 d, 14 d, and 28 d by GC-MS, contained aldehydes (10), esters (10), acids (2), alcohols (14), ketones (2), terpenes (4), and others (5), constituting seven kinds of compounds. There were many changes between single sample groups at different time points, and at the same time point between different sample groups, mainly in terms of flavor ratios and inconsistencies in composition. Qian et al. [51] mainly detected terpenoids, aldehydes, alcohols, esters, and ketones from blueberry, which is different from the results of this study, and may be related to blueberry varieties. Compared with the CK group, HT treatments effectively delayed the degradation of esters in aromatic components and inhibited the formation of alcohols in blueberry fruit during storage. In previous studies, increased alcohols and decreased esters in blueberry may accelerate blueberry ripening [52]. A total of 47 components were detected (Appendix A, Table A2). All the treatment groups contained ethyl acetate, methyl acetate, methyl isopentate, n-hexanal, ethanol, nonanaldehyde, n-hexanol, (+)-limonene, linalool, 4-terpenol, and methylheptaneone throughout the storage period, with the highest content obtained from ethyl acetate and (+)-limonene. It has been reported that ethyl acetate was one of the major volatile components in rabbiteye blueberries [53]. At 14 d of storage, compared with the CK group, HT2 treatment increased the ethyl acetate content 75-fold (*p* < 0.05), but with a significant decrease in (+)-limonene content. However, no significant difference was found in the (+)-limonene content of blueberries between HT and CK groups (*p* > 0.05) on 28 d, and HT2 exhibited a small increase.

Hierarchical clustering analysis was used on the 47 components detected during storage using Origin Pro 2021b (Figure 6d), which showed that the single application of TKL coating (HT1) had no significant effect on most volatile flavor concentrations and classes during storage compared with CK. Moreover, the TKL coating combined with a mild heat-shock (45 °C or 65 °C) showed the best performance for refrigerated blueberries after 14 d, retaining some degree of similarity to fresh fruits in agreement with electronic nose response signals. In conclusion, the combination of heat-shock and TKL edible coating treatments was more conducive to preserving the flavor of blueberries harvested during the compared with CK and HT1.

#### 3.7.2. Sensory Evaluation

The flavor of blueberries is determined by taste and odor components, predominantly sugars, acids, and aromatic compounds. If TKL edible coatings are to be used as natural bio preservatives on blueberries, then they should not have deleterious effects on the sensory attributes of the fruits. The fuzzy comprehensive sensory evaluation performed on heat-shocked and un-heat-shocked blueberries found that the use of edible coatings did not introduce deleterious effects to the sensory acceptability of blueberries harvested (Figure 7). The most desirable qualities were blue color; crisp, fragrant, and juicy flesh; suitable acid levels; and sweet flavor. Sensory scores did not differ significantly between treatment groups during blueberry storage time but were higher in HT groups than in CK groups at various time points, with the best response among them being in HT2. The HT2 treatment was ideal for maintaining fruit quality, as the scores were 15.31% and 53.44% higher than those of the control on 14 d and 28 d, respectively. However, when combined treatment was applied at higher temperatures (65 °C), negative effects were observed, namely increased fungal decay, softer fruits, greater nutrient loss, increased weight loss, and berry color darkening, compared with the controls in our study. These results are consistent with those of previous studies on other berries [20,54,55]. A possible explanation is that the application of improper heat-shock temperature damaged the fruit tissue and stimulated fruit respiration. For this reason, there is a need to determine the most effective combination treatment of heat-shock and TKL edible coating for postharvest blueberry fruits that will produce the desired effect (decay control, fungal control, and fluid loss control) without damaging the commodity.

## 4. Conclusions

Compared with other concentrations of TKL edible coating (20, 40, 80, and 100 mg/L), 60 mg/L TKL edible coating has good antifungal, antioxidant and preservation effects. We also compared the effects of different heat-shock temperatures (25, 35, 45, 55 and 65 °C) on the quality of blueberry and found that the quality of blueberry treated by heat-shock at 25 °C, 45 °C and 65 °C was significantly different. Therefore, the combination of TKL60 and HT treatment (25, 45 and 65 °C) was used to improve the storage quality of blueberry. This study found that the application of thermal treatment following coating under suitable conditions reduced weight loss, anthocyanin loss, *L*-Vc loss and fruit skin color changes during storage. It also delayed the senescence process of blueberry fruit, mainly by inhibiting the increase of respiratory intensity and ethylene release rate and delaying the decrease of TA and SSC content. In addition, we confirmed that the 45 °C heat-shock and edible coating combined treatment exhibited the most favorable benefits on edible quality including texture, taste, and aroma, and thus has great potential for application in fruit preservation.

## Figures and Tables

**Figure 1 foods-12-00789-f001:**
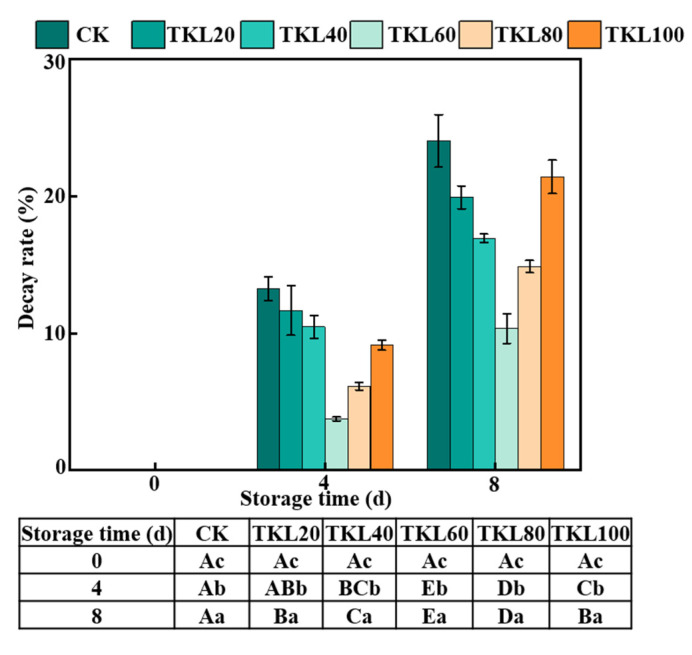
Decay rate of TKL in blueberries during storage times at 25 °C (*n* = 3). Different uppercase letters indicate significant difference between the groups (*p* < 0.05), and different lowercase letters indicate significant difference within the groups (*p* < 0.05) (same as below).

**Figure 2 foods-12-00789-f002:**
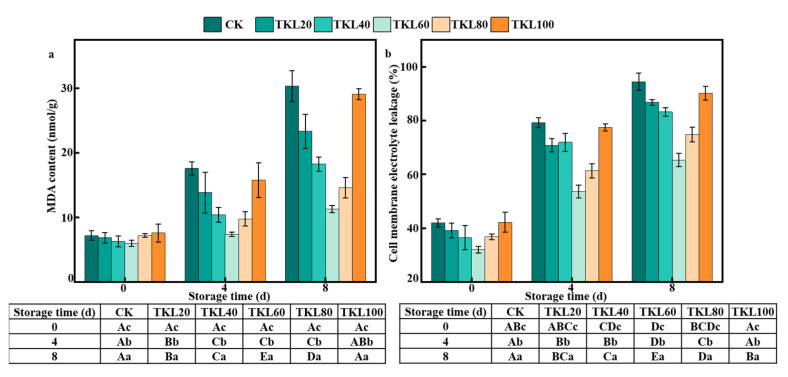
(**a**) MDA content, (**b**) Cell membrane electrolyte leakage of TKL in blueberries during storage times at 25 °C (*n* = 3). Different uppercase letters indicate significant difference between the groups (*p* < 0.05), and different lowercase letters indicate significant difference within the groups (*p* < 0.05).

**Figure 3 foods-12-00789-f003:**
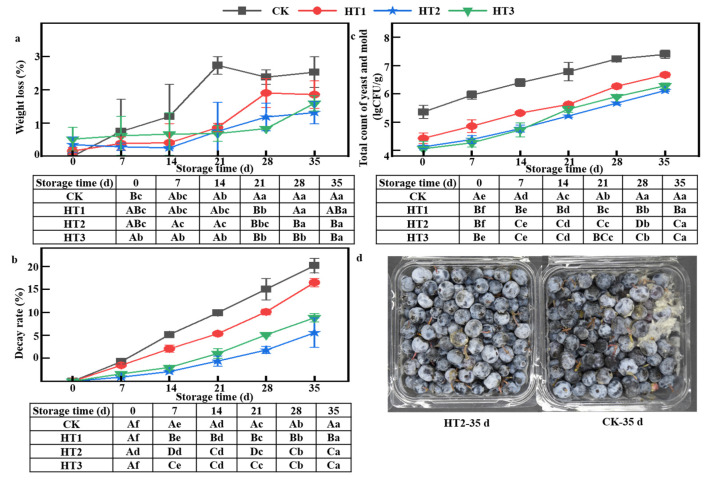
(**a**) Weight loss. (**b**) Decay rate. (**c**) Total count of yeast and mold of HT in blueberries during storage times at 2 °C (*n* = 6). (**d**) Images of freshness preservation of HT in blueberries on the 35th day at 2 °C. Different uppercase letters indicate significant difference between the groups (*p* < 0.05), and different lowercase letters indicate significant difference within the groups (*p* < 0.05).

**Figure 4 foods-12-00789-f004:**
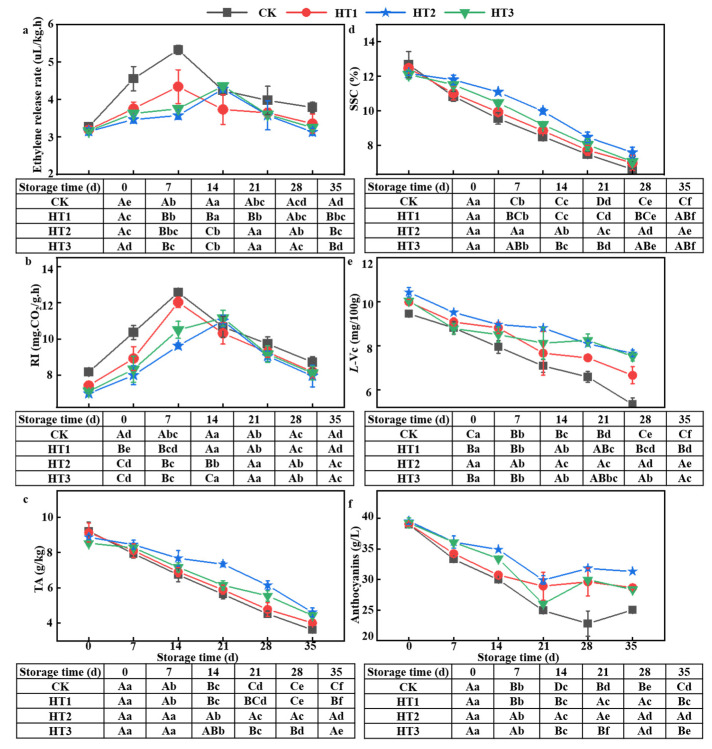
Physiological and biochemical characterization of HT in blueberries during storage times at 2 °C (*n* = 6); (**a**–**f**) represent ethylene release rate, RI, TA, SSC, *L*-Vc and anthocyanins, respectively. Different uppercase letters indicate significant difference between the groups (*p* < 0.05), and different lowercase letters indicate significant difference within the groups (*p* < 0.05).

**Figure 5 foods-12-00789-f005:**
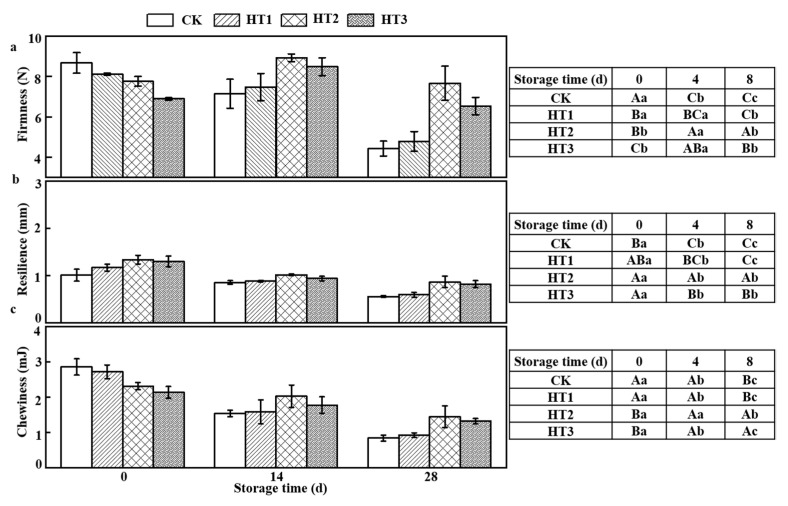
TPA of HT in blueberries during storage times at 2 °C (*n* = 3); (**a**–**c**) firmness, resilience, chewiness, respectively. Different uppercase letters indicate significant difference between the groups (*p* < 0.05), and different lowercase letters indicate significant difference within the groups (*p* < 0.05).

**Figure 6 foods-12-00789-f006:**
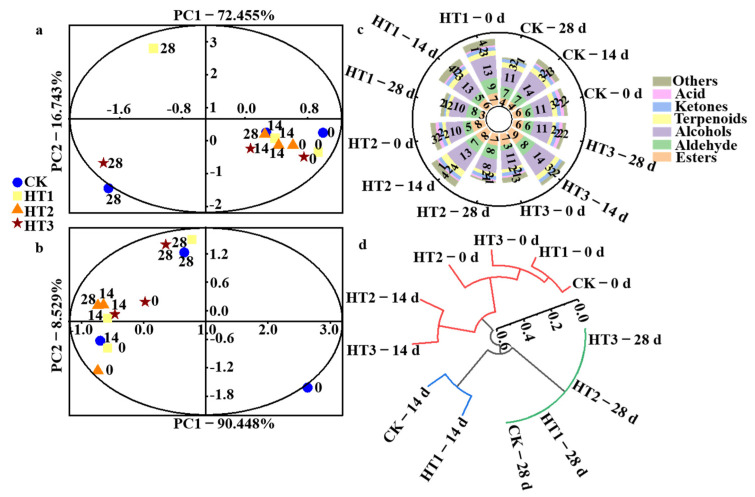
Flavor characterizations of HT in blueberries during storage times at 2 °C (*n* = 3); (**a**,**b**) represent PAC of taste and odor fingerprint profile, respectively; (**c**) represents aroma components; (**d**) represents hierarchical clustering analysis.

**Figure 7 foods-12-00789-f007:**
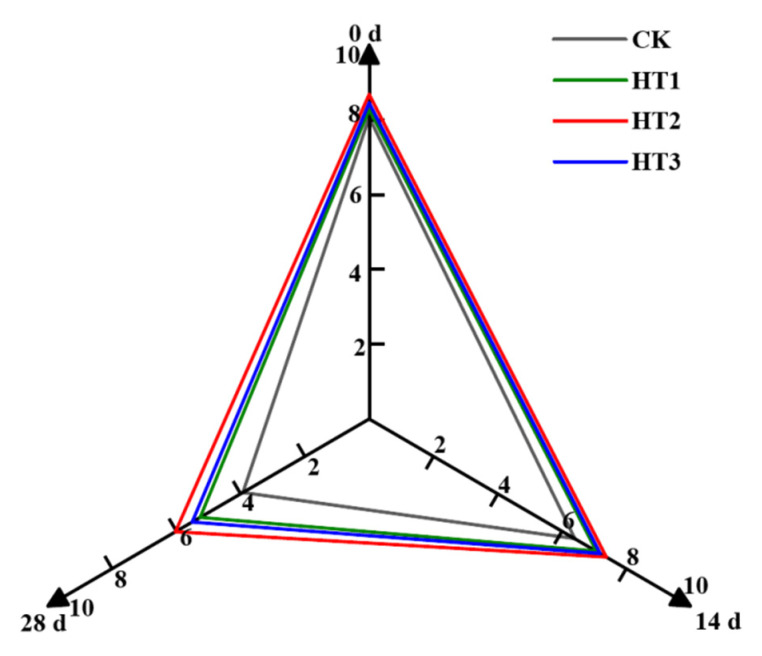
Sensory evaluation of HT in blueberries during storage times at 2 °C (*n* = 3).

**Table 1 foods-12-00789-t001:** Sensory evaluation criteria of blueberry.

Indicators	Scoring Criteria	Score
Smell	The flavor is excellent, the sugar acid ratio is very harmonious, and has a rich fragrance	9
	Flavor coordination, sugar and acid ratio coordination, blueberry aroma is normal	7
	The flavor is ok, the sugar–acid ratio is not harmonious, the blueberry aroma is light	5
	The flavor is not harmonious, the sugar acid ratio is not harmonious, has a very sour taste	3
	The flavor is poor and almost inedible	1
Taste	The pulp is chewy and juicy	9
	Pulp chewing better, juice general	7
	Poor chewing, less juice	5
	Chewing is very poor, very little juice	3
	Non-chewable and almost inedible	1
Tissue state	The peel has good elasticity and moderate hardness	9
	The peel is elastic and slightly soft	7
	Peel elasticity is poor, easy to break, slanting and soft	5
	Poor peel elasticity, easy to crack, soft, poor degree of skin and flesh	3
	The skin is broken and almost inedible	1
Color	Blueberry powder is complete and bright black	9
	Blueberry powder is more complete, black	7
	Blueberry powder is pink and dark red	5
	Blueberry powder is less, showing burgundy	3
	No blueberry powder, red	1

**Table 2 foods-12-00789-t002:** Antifungal activity and antioxidant activity of TKL compound coating.

TKL Composite Coating Concentration (mg/L)	Antifungal Activity (%)			Antioxidant Activity (%)
	*Penicillium* *expansum*	*Alternaria nees*	*Botrytis cinerea*	
20	14.6 ± 3.9 ^b^	1.1 ± 1.0 ^c^	5.8 ± 1.9 ^c^	33.1 ± 0.4 ^b^
40	22.2 ± 5.8 ^b^	4.7 ± 3.1 ^bc^	55.8 ± 5.1 ^b^	34.1 ± 1.4 ^b^
60	37.3 ± 8.8 ^a^	10.9 ± 1.7 ^ab^	63.5 ± 6.5 ^ab^	47.9 ± 1.6 ^a^
80	40.2 ± 10.5 ^a^	12.4 ± 6.4 ^a^	67.3 ± 6.4 ^a^	49.3 ± 5.6 ^a^
100	50.3 ± 11.7 ^a^	16.7 ± 6.4 ^a^	71.7 ± 8.5 ^a^	54.6 ± 7.7 ^a^

Different letters indicate significant difference between the groups (*p* < 0.05).

**Table 3 foods-12-00789-t003:** Effect of heat-shock temperature (25, 35, 45, 55 and 65 °C) on blueberry quality during storage times at 25 °C (*n* = 3).

Index	Storage Time (d)	CK	25 °C	35 °C	45 °C	55 °C	65 °C
Decay rate (%)	0	0.0 ± 0.0 ^Ac^	0.0 ± 0.0 ^Ac^	0.0 ± 0.0 ^Ac^	0.0 ± 0.0 ^Ac^	0.0 ± 0.0 ^Ac^	0.0 ± 0.0 ^Ac^
	4	13.6 ± 0.6 ^Ab^	11.9 ± 0.3 ^Bb^	9.8 ± 0.5 ^Cb^	9.3 ± 0.1 ^Cb^	8.2 ± 0.3 ^Db^	7.5 ± 0.9 ^Db^
	8	22.2 ± 0.6 ^Aa^	20.7 ± 0.3 ^Ba^	16.8 ± 1.1 ^Ca^	15.4 ± 0.8 ^Ca^	13.1 ± 0.7 ^Da^	12.2 ± 1.1 ^Da^
MDA content (nmol/g)	0	6.7 ± 0.2 ^Ac^	6.7 ± 0.3 ^Ac^	6.1 ± 0.8 ^Ac^	6.3 ± 0.3 ^Ac^	6.1 ± 0.7 ^Ac^	6.2 ± 1.2 ^Ac^
	4	18.2 ± 0.5 ^Ab^	17.6 ± 0.5 ^Ab^	14.8 ± 0.8 ^Bb^	13.0 ± 1.2 ^Bb^	11.9 ± 0.3 ^Cb^	10.2 ± 1.5 ^Cb^
	8	29.3 ± 0.9 ^Aa^	28.2 ± 0.3 ^Aa^	24.2 ± 2.3 ^Ba^	22.7 ± 0.6 ^Ba^	21.0 ± 0.9 ^Ca^	19.0 ± 1.7 ^Ca^
Cell membrane electrolyte leakage (%)	0	46.8 ± 1.9 ^Ac^	45.4 ± 2.4 ^Ac^	44.4 ± 0.8 ^Ac^	44.9 ± 2.2 ^Ac^	45.8 ± 1.3 ^Ac^	45.9 ± 1.8 ^Ac^
	4	72.9 ± 0.9 ^Ab^	68.6 ± 2.3 ^Bb^	64.9 ± 1.5 ^Cb^	62.7 ± 0.6 ^Cb^	59.3 ± 1.9 ^Db^	56.9 ± 2.9 ^Db^
	8	96.2 ± 2.7 ^Aa^	90.6 ± 2.1 ^Ba^	85.5 ± 2.2 ^Ca^	83.9 ± 3.1 ^Ca^	78.9 ± 1.3 ^Da^	76.5 ± 2.4 ^Da^

Different uppercase letters indicate significant difference between the groups (*p* < 0.05), and different lowercase letters indicate significant difference within the groups (*p* < 0.05).

**Table 4 foods-12-00789-t004:** Color of HT in blueberries during storage times at 2 °C.

Index	Group	Storage Time (d)		
		0	14	28
*L**	CK	14.29 ± 2.15 ^Ba^	10.59 ± 0.92 ^Bb^	9.05 ± 0.61 ^Cb^
	HT1	14.88 ± 1.22 ^Ba^	11 ± 0.59 ^Bb^	9.57 ± 0.48 ^BCc^
	HT2	16.58 ± 1.41 ^Aa^	12.86 ± 1.76 ^Ab^	10.79 ± 0.64 ^Ac^
	HT3	15.07 ± 0.33 ^ABa^	11.76 ± 0.44 ^ABb^	10.12 ± 0.4 ^Bc^
*a**	CK	0.4 ± 0.15 ^Ac^	0.71 ± 0.09 ^Ab^	1.46 ± 0.17 ^Aa^
	HT1	0.37 ± 0.44 ^Ab^	0.6 ± 0.1 ^Ab^	1.25 ± 0.33 ^Aa^
	HT2	0.26 ± 0.15 ^Ab^	0.44 ± 0.4 ^Aab^	0.86 ± 0.64 ^Aa^
	HT3	0.33 ± 0.3 ^Aa^	0.5 ± 0.3 ^Aa^	0.91 ± 0.78 ^Aa^
*b**	CK	−2.54 ± 0.75 ^Ab^	−1.7 ± 0.62 ^Aa^	−1.12 ± 0.24 ^Aa^
	HT1	−2.99 ± 0.72 ^ABc^	−2.03 ± 0.72 ^Ab^	−1.29 ± 0.14 ^Aa^
	HT2	−3.7 ± 0.32 ^Bc^	−2.69 ± 0.34 ^Bb^	−1.66 ± 0.52 ^Ba^
	HT3	−3.2 ± 0.63 ^ABc^	−2.35 ± 0.5 ^ABb^	−1.39 ± 0.18 ^ABa^

Different uppercase letters indicate significant difference between the groups (*p* < 0.05), and different lowercase letters indicate significant difference within the groups (*p* < 0.05).

## Data Availability

Not applicable.

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
