# Peer review of "Use of Heat-Shock and Edible Coating to Improve the Postharvest Preservation of Blueberries"

_foods, 2023, doi:10.3390/foods12040789_

Round 1
Reviewer 1 Report
The manuscript presents a very detailed and comprehensive investigation of effects of thymol-containing edible coatings and heat treatments, separate and in combination, on the quality of blueberries. However, the methodology of the study and its description rise some questions and concerns that have to be answered.
My major concern is about the methodology of heat treatment. Usually, application of heat treatments to fresh produce is accompanied by means to reduce the produce desiccation (e.g., plastic wrapping or air humidification). However, in the present case, the equipment used for heat treatment was electric blast drying oven (l. 187 and 199-200) specially designed for drying; the exposure time was relatively long (1 h) and no any desiccation protection is mentioned. How authors could ensure that this treatment exerted no damage to produce? Yang et al. (HORTSCIENCE 54(12):2231–2239. 2019) showed that even preharvest exposure to temperatures above 40°C (although for somewhat longer time) caused damage (e.g. shriveling) to blueberries. There is no reason to expect that the detached berries are less desiccation-sensitive, opposite. The authors calculate weight loss related to initial storage point (i.e. post-treatment) but ignore the weight loss DURING THE TREATMENT in the drying oven. Without these data, the presented information (e.g. weight loss reduction by the treatment) can be incorrect and misleading. IT IS ESSENTIAL to provide this information, even if it requires running an additional trial, and to consider it in the results interpretation.
Some additional comments and recommendations:
As specified in l. 39, the blueberry harvest season is always rainy, so there is no reason to specify "after rainy season" in the title, to discuss the detrimental rain effects on the blueberry quality storability, and associate high water content in the fruit with rain (l. 39-44). Is water content lower in blueberries grown under protected cultivation conditions? It seems sufficient just to mark that blueberries are perishable.
In the title, not "preservation of postharvest blueberries" but "postharvest preservation of blueberries".
In the abstract, the "TLK60 treatments" is used without any explanation. Some journals demand no abbreviations in the abstract. At least, specify "TLK60 composite coating".
In l. 43-46, the use of some postharvest treatments is justified by references, while others not. Be consistent.
Why increase of antioxidant properties negatively affects the fruit quality (l. 55-56)?
What does it mean "commodity rate" (l. 75) and how "preservation effect" (l. 159) is determined?
"Different concentrations of TKL composite coatings" (l. 152-153), from 20 to 100 mg/L – milligrams of what? Thymol?
Spraying 2.5 mL coating per gram (of berries?) (l. 157) gives huge amounts of liquid, e.g. 250 mL per 100-g blueberry sample. Is it correct? How could it be applied?
What substrate (e.g., whole fruit?) was examined by the conductivity meter (l. 178-180) and how the measurement was done and results calculated and interpreted? What did it mean?
The coated fruit was "air dried before harvest" (l. 197). Does it mean that the coating was done preharvest? This is not explained clearly. What does it mean that the sprayer was "farm-oriented" (l. 195)?
Showing one individual blueberries in Fig. 1 and 3 is not very meaningful because of the great variability of the fruit in each sample. I recommend removing them. On the other hand, the photographs of punnetts in Fig. 3 should be enlarged (2 rows, 2 punnetts in each row) to make them better visible.
The difference between HT2 (45°C) and HT3 (65°C) treatments is in most cases not significant (e.g. in Fig. 3). Therefore, the conclusion about the advantage of 45 over 65°C (e.g. in the abstract) may be "softened". For example, instead of the sentence in l. 23-24, it might be written something like, "The heat shock treatments were effective in maintaining the quality of blueberries, with certain advantage of 45°C over 65 °C".
The authors often misuse the word "however", in particular in l. 23, 47, maybe 54.
Author Response
Point 1: My major concern is about the methodology of heat treatment. Usually, application of heat treatments to fresh produce is accompanied by means to reduce the produce desiccation (e.g., plastic wrapping or air humidification). However, in the present case, the equipment used for heat treatment was electric blast drying oven (l. 187 and 199-200) specially designed for drying; the exposure time was relatively long (1 h) and no any desiccation protection is mentioned. How authors could ensure that this treatment exerted no damage to produce? Yang et al. (HORTSCIENCE 54(12):2231–2239. 2019) showed that even preharvest exposure to temperatures above 40°C (although for somewhat longer time) caused damage (e.g. shriveling) to blueberries. There is no reason to expect that the detached berries are less desiccation-sensitive, opposite. The authors calculate weight loss related to initial storage point (i.e. post-treatment) but ignore the weight loss DURING THE TREATMENT in the drying oven. Without these data, the presented information (e.g. weight loss reduction by the treatment) can be incorrect and misleading. IT IS ESSENTIAL to provide this information, even if it requires running an additional trial, and to consider it in the results interpretation.
Response 1: Thanks, your opinion is very pertinent. The results of Yang et al. showed that (HORTSCIENCE 54(12):2231–2239. 2019) berries at the blue stage, had little to no damage within 3 to 4 hours at 42 to 46 °C. Like many fruits, blueberries possess inherent qualities such as a waxy cuticle that provides natural protection against heat damage. In our experiment, the blueberry fruit was treated with TKL coating before heat treatment, which was equivalent to wearing an extra dress to the fruit in advance and improving the heat resistance of the fruit. Figure 3a records the weight loss during the treatment in the drying oven. That is, the weight loss of 0 days of storage.
Some additional comments and recommendations:
Point 2: As specified in l. 39, the blueberry harvest season is always rainy, so there is no reason to specify "after rainy season" in the title, to discuss the detrimental rain effects on the blueberry quality storability, and associate high water content in the fruit with rain (l. 39-44). Is water content lower in blueberries grown under protected cultivation conditions? It seems sufficient just to mark that blueberries are perishable.
Response 2: Revised. Thanks.
Point 3: In the title, not "preservation of postharvest blueberries" but "postharvest preservation of blueberries".
Response 3: Revised. Thanks.
Point 4: In the abstract, the "TLK60 treatments" is used without any explanation. Some journals demand no abbreviations in the abstract. At least, specify "TLK60 composite coating".
Response 4: Revised. Thanks.
Point 5: In l. 43-46, the use of some postharvest treatments is justified by references, while others not. Be consistent.
Response 5: Revised. Thanks.
Point 6: Why increase of antioxidant properties negatively affects the fruit quality (l. 55-56)?
Response 6: Revised. Thanks. External stimulation can induce fruit defense, including the increase of defense enzyme activity. At the same time, it will also lead to a decrease in fruit quality. (10.1016/j.postharvbio.2022.112020). Gabriela et al. also reported that UV-C would increase the loss of fruit weight. (10.1016/j.heliyon.2021.e07190)
Point 7: What does it mean "commodity rate" (l. 75) and how "preservation effect" (l. 159) is determined?
Response 7: Revised. Thanks. FAN et al. calculated the commodity rate as: Blueberries were then sorted into 4 categories: (1) marketable: unblemished fruit; (2) shriveled: any fruit with visible outer skin wrinkling; (3) split: any fruit with a visible fracture in its outer skin, and (4) decay: any fruit with visible mold growth. Fruit in each category were weighed and expressed as a percentage of the total weight.
The original preservation effect is shown by the decay of the fruit surface, but now the photo has been deleted according to your teacher 's opinion.
Point 8: "Different concentrations of TKL composite coatings" (l. 152-153), from 20 to 100 mg/L – milligrams of what? Thymol?
Response 8: Thanks. Yes, It can be seen in 119-121 lines.
Point 9: Spraying 2.5 mL coating per gram (of berries?) (l. 157) gives huge amounts of liquid, e.g. 250 mL per 100-g blueberry sample. Is it correct? How could it be applied?
Response 9: Thanks. For TKL spray treatments, fruit were sprayed until they were completely moistened.Mist spraying was carried out using an electric sprayer with centrifugal nozzle, where the nozzle diameter is 0.5 mm.The actual rate of application of fruit sprays was 250 mL/100 g.The final result is summarized based on multiple times of experimental confirmation.
Point 10: What substrate (e.g., whole fruit?) was examined by the conductivity meter (l. 178-180) and how the measurement was done and results calculated and interpreted? What did it mean?
Response 10: Revised. Thanks. The relative electrical conductivity (REC) was determined using a conductivity meter (Model DDS-11a, Shanghai Scientific Instruments, Shanghai, China). Take 0.20 g peel (5 mm × 5 mm) into a conical flask and add 30 mL distilled water, placed in a vacuum dryer for 10 min to extract intercellular air. Slowly put into the air, water penetrates into the intercellular space, the leaves become transparent, and the intra-cellular solute is easy to exude. The conical flask was taken out and oscillated every few minutes. The conductivity L1 was measured after being kept at room temperature for 30 min. Then the plugged conical flask was transferred into boiling water for 20 min. The conductivity L2 was measured after cooling to room temperature. Each test sample was assayed in triplicate and the experiment was repeated three times. The REC was calculated as follows:REC(%)=L1/L2×100 (I184-194)
Point 11: The coated fruit was "air dried before harvest" (l. 197). Does it mean that the coating was done preharvest? This is not explained clearly. What does it mean that the sprayer was "farm-oriented" (l. 195)?
Response 11: Revised. Thanks. The TKL film was evenly coated on the surface of ripe fruits using agricultural atomizing sprayer (Model 280, China Weifang Motong Machinery Manufacturing Co., Ltd), after the coating solution had dried completely, blueberry fruit were picked. (I209-211)
Point 12: Showing one individual blueberries in Fig. 1 and 3 is not very meaningful because of the great variability of the fruit in each sample. I recommend removing them. On the other hand, the photographs of punnetts in Fig. 3 should be enlarged (2 rows, 2 punnetts in each row) to make them better visible.
Response 12: Revised. Thanks.
Point 13: The difference between HT2 (45°C) and HT3 (65°C) treatments is in most cases not significant (e.g. in Fig. 3). Therefore, the conclusion about the advantage of 45 over 65°C (e.g. in the abstract) may be "softened". For example, instead of the sentence in l. 23-24, it might be written something like, "The heat shock treatments were effective in maintaining the quality of blueberries, with certain advantage of 45°C over 65 °C".
Response 13: Revised. Thanks.
Point 14: The authors often misuse the word "however", in particular in l. 23, 47, maybe 54.
Response 14: Revised. Thanks.

Reviewer 2 Report
The manuscript about the effect of heat treatment and edible coating on post-harvest quality of blueberries is a work with mainly practical application and the manuscript is average written. Some issues that I point out are below.
The title is not totally representative of the study. The addition of ‘after rainy season’ prepares the author to read about a study with at least 2 different harvesting periods with different climate conditions just before the harvest. Otherwise since is normal to cultivate blueberries under rainy conditions as the authors reported in the introduction, the title must be changed.
- Keywords that already exist in the title must be replaced.
- More information is required in the statistical analysis section. Authors just reported in another section that they used 3 replicates, but what about the number of fruits per replication as well as the experimental design?
- In lines 207-208 authors reported the criteria for the ending of the marketable period. Documentation is required otherwise it seems to be an arbitrary decision.
- The captions of tables must be more detailed for the reader.
Author Response
Point 1: The title is not totally representative of the study. The addition of ‘after rainy season’ prepares the author to read about a study with at least 2 different harvesting periods with different climate conditions just before the harvest. Otherwise since is normal to cultivate blueberries under rainy conditions as the authors reported in the introduction, the title must be changed.
Response 1: Revised. Thanks.
Point 2: Keywords that already exist in the title must be replaced.
Response 2: Revised. Thanks.
Point 3: More information is required in the statistical analysis section. Authors just reported in another section that they used 3 replicates, but what about the number of fruits per replication as well as the experimental design?
响应 3:已修改。谢谢。你的意见很中肯。
第 4 点:在第 207-208 行中,作者报告了适售期结束的标准。需要文件,否则这似乎是一个武断的决定。
响应 4:已修改。谢谢。你的意见很中肯
(10.1016/j.foodchem.2022.134227)。
第 5 点:表格的标题必须为读者提供更详细的信息。
响应 5:已修改。谢谢。你的意见很中肯。

Reviewer 3 Report
Review foods-2063167, 14.01.23
Use of heat shock and edible coating to enhance the preservation of postharvest blueberries after rainy season
The title is good, illustrative, but instead of the word "enhance" I would use improve
Line 29: assessment → evaluations or examination
Line 30: treated → treated with heat shock …. did not show a large placement of
Line 38: because of → due to
Line 39: hot and rainy → should be explained where because in my country is usually hot and dry
Materials and Methods
First you should write what plant material you have and how you obtain fruits with coordinates of orchard and next you should list the materials used for coatings production in the same order like you present results. I mean you should group them. If you write at the beginning number of treatment then the whole paper will be easier to follow.
2.4.1. How was obtain uniformly sprayed fruits? How to avoid sometimes double layer has not been applied to the fruit surface?
2.4.2. What was the ‘n’?
2.6.2. You have measured many parameters that are an indirect assessment of the intensity of respiration or the result of this respiration by changing the chemical composition, and why you did not measure the production of ethylene, which is one of the better parameters for assessing the degree of ripeness of climacteric and nonclimacteric fruits? However, I do not really understand why you measured soluble protein content?
6) I do not know why you measured the content of pelargonidins in fruits? Blueberries do not contain pelargonidin. See Journal of Berry Research 2 (2012) 179–189 DOI:10.3233/JBR-2012-038
7) how many fruits were measured?
2.6.4.
2) needs more details about scale used for evaluation. There were 5 parameters evaluation and but final result is the only one.
Results
Table 1.
Unify number digits after coma. In this case one digit is enough.
Lines 344-345: What bacteria’s can caused the decay?
Figure 1.
0 day is Ac? Means that is no difference between 0 and 4 and 8 day in CK? You cant make ANOVA if value is 0! It should be provided a new analyze.
3.3.
The first sentence is useless.
3.4.
You are not citing the table 2.
Table 2.
You must also round the data to the first decimal place.
Figure 3. The bars should be at least twice bigger especially that the figure discussed the data discussed in a few paragraph.
3.5.3. Again bacteria – yeast and mold?
3.6.
If you do not place digital data, you should make the charts legible so that you can read the approximate values from the chart. Currently, from the chart no4 is very difficult, and actually impossible. Which makes it difficult to assess the discussion of this part of the work.
Opinions whether blueberry is a climacteric fruit or not the discussion has been going on for a long time. However, the course of breathing intensity is like in climacteric fruits, so it is not surprising that factors affecting the rate of breathing shape the breathing curve. Part of RI should be discussed through the prism of fruit ripeness. WHAT accelerates it and what delays it is crucial in this aspect. All the more so because sensitivity to diseases is also closely related to the degree of maturity. Why does temperature shock delay maturation? Does it have something to do with enzymatic activity or is the cause different? How TA and TSS are associated with maturity?
3.6.6. How anthocyanins influence the flavor, please provide some citation.
The same remark as above. There are on figure 5 significant differences but not visible.
3.72. Pure sugar does not have any smell.
4.0.
This part is weak. Needs to be revise. Please provide all achievements separately now this big experiment is summarized too poorly.
Author Response
Point 1: Use of heat shock and edible coating to enhance the preservation of postharvest blueberries after rainy season. The title is good, illustrative, but instead of the word "enhance" I would use improve
Response 1: Revised. Thanks, your opinion is very pertinent.
Point 2: Line 29: assessment → evaluations or examination
Response 2: Revised. Thanks.
Point 3: Line 30: treated → treated with heat shock …. did not show a large placement of
Response 3: Revised. Thanks.
Point 4:: Line 38: because of → due to
Response 4: Revised. Thanks.
Point 5: Line 39: hot and rainy → should be explained where because in my country is usually hot and dry.
Response 5: Revised. Thanks, your opinion is very pertinent.
Materials and Methods
Point6: First you should write what plant material you have and how you obtain fruits with coordinates of orchard and next you should list the materials used for coatings production in the same order like you present results. I mean you should group them. If you write at the beginning number of treatment then the whole paper will be easier to follow.
Response 6: Revised. Thanks. Your opinion is very pertinent.
Point 7: 2.4.1. How was obtain uniformly sprayed fruits? How to avoid sometimes double layer has not been applied to the fruit surface?
Response 7: Revised. Thanks. Your opinion is very pertinent. For TKL spray treatments, the fruits were sprayed until they were completely wet, using an electric sprayer with a centrifugal nozzle with a nozzle diameter of 0.5 mm. The actual application rate of fruit sprays was 250 ml/100 g. Several pre-tests have confirmed that at the above dosage it is possible to ensure that most of the fruit is completely moistened and that a more uniform film is formed on the surface of the fruit after drying.
Point 8: 2.4.2. What was the ‘n’?
Response 8: Thank you very much for your review. Where n is the total number of examined fruits in each sample.
Point 9: 2.6.2. You have measured many parameters that are an indirect assessment of the intensity of respiration or the result of this respiration by changing the chemical composition, and why you did not measure the production of ethylene, which is one of the better parameters for assessing the degree of ripeness of climacteric and nonclimacteric fruits? However, I do not really understand why you measured soluble protein content?
Response 9: Revised. Thanks. Your opinion is very pertinent.
Point 10: 6) I do not know why you measured the content of pelargonidins in fruits? Blueberries do not contain pelargonidin. See Journal of Berry Research 2 (2012) 179–189 DOI:10.3233/JBR-2012-038
Response 10: Thank you very much for your review. The method refers to the Chinese standard-high performance liquid chromatography in the determination of anthocyanins in plant-derived foods and makes appropriate modifications. This method is a standard curve established by using a mixed standard, so the method contains geranium pigment. But the total anthocyanin content in our results report is without geranium pigment (I280-281).
Point 11: 7) how many fruits were measured?
Response 11: Revised. Thanks. Each treatment selected 50 fruits for measured.
Point 12: 2.6.4.2) needs more details about scale used for evaluation. There were 5 parameters evaluation and but final result is the only one.
Response 12: Revised. Thanks.
Results
Point 13: Table 1. Unify number digits after coma. In this case one digit is enough.
Response 13: Revised. Thanks.
Point 14: Lines 344-345: What bacteria’s can caused the decay?
Response 14: Thanks. Your opinion is very pertinent. It is reported that Penicillium, Alternaria nees, and Botrytis cinerea are common pathogens of postharvest decay of blueberry. ( 10.11937/bfyy.20182597、10.16429/j.1009-7848.2020.02.033)
Point 15: Figure 1.0 day is Ac? Means that is no difference between 0 and 4 and 8 day in CK? You cant make ANOVA if value is 0! It should be provided a new analyze.
Response 15: Thanks. Your opinion is very pertinent. A in Ac indicates that there is no significant difference between different treatment groups, and c indicates the difference of the same treatment at different times. The decay rate of CK group fruit at 4 days and 8 days was expressed as b and a respectively, which was different from c at 0 days, indicating that the decay rate of CK group fruit at different times was significantly different. Data processing and labeling methods refer to the research of Ding Jie et al. (10.1016/j.foodchem.2022.134227)
Point 16: 3.3.The first sentence is useless.
Response 16: Revised. Thanks.
Point 17: 3.4.You are not citing the table 2.
Response 17: Revised. Thanks.
Point 18: Table 2.You must also round the data to the first decimal place.
Response 18: Revised. Thanks.
Point 19: Figure 3. The bars should be at least twice bigger especially that the figure discussed the data discussed in a few paragraph.
Response 19: Revised. Thanks.
Point 20: 3.5.3. Again bacteria – yeast and mold?
Response20: Yes, yeast and mold. Revised. Thanks.
Point 21: 3.6.If you do not place digital data, you should make the charts legible so that you can read the approximate values from the chart. Currently, from the chart no4 is very difficult, and actually impossible. Which makes it difficult to assess the discussion of this part of the work.
Response 21: Revised. Thanks. Your opinion is very pertinent.
Point 22: Opinions whether blueberry is a climacteric fruit or not the discussion has been going on for a long time. However, the course of breathing intensity is like in climacteric fruits, so it is not surprising that factors affecting the rate of breathing shape the breathing curve. Part of RI should be discussed through the prism of fruit ripeness. WHAT accelerates it and what delays it is crucial in this aspect. All the more so because sensitivity to diseases is also closely related to the degree of maturity. Why does temperature shock delay maturation? Does it have something to do with enzymatic activity or is the cause different? How TA and TSS are associated with maturity?
Response 22: Thanks. Your opinion is very pertinent. Fruit senescence accelerates respiratory intensity. The molecular membrane barrier formed by TKL composite coating can delay the increase of respiratory intensity. The reasons why temperature shock delays fruit ripening are as follows: (1) The physiological and biochemical changes in heat-treated fruit are more advanced in some ripening characteristics than non-treated fruit, thus maintaining the fruit quality for a longer period during storage and shelf-life. (2) Hot shock caused the melting of wax, which partially or entirely sealed natural openings in the epidermis, prevented the invasion of the pathogen and inactivated quiescent/latent infections of pathogen, and then reduced disease incidence. Studies have shown that the decrease of soluble sugar and titratable acid in blueberry fruit will shorten the shelf life of fruit. (10.1016/j.ifset.2017.04.007) \ (10.1016/j.lwt.2016.12.056.) (10.1016/j.postharvbio.2013.08.001)\ (10.1016/j.foodchem.2022.135187)\ (10.1016/j.foodchem.2022.135187)
Point 23: 3.6.6. How anthocyanins influence the flavor, please provide some citation.
Response 23: Thanks. Your opinion is very pertinent. Anthocyanins may affect the synthesis of some amino acids. Amino acids are also important nutrients in the fruit, some of which are related to the synthesis of fruit flavor substances. The amino acids involved in the synthesis of flavor substances are called delicious amino acids. According to the human sense of taste, delicious amino acids can be divided into flavor amino acids (glutamic acid and aspartic acid), sweet amino acids (alanine, glycine, serine, threonine, and proline) and bitter amino acids (valine, isoleucine, leucine, arginine, phenylalanine (Phe), histidine, and methionine). Phe. As an aromatic amino acid, Phe is a precursor to the synthesis of anthocyanins and flavonoids, which play an important role in their biosynthesis process. (10.3390/foods11243965)
Point 24: The same remark as above. There are on figure 5 significant differences but not visible.
Response 24: Revised. Thanks. Your opinion is very pertinent.
Point 25: 3.72. Pure sugar does not have any smell.
Response 25: Thank you very much for your review. Yes, Pure sugar does not have any smell. But blueberry fruit has a unique blueberry flavor, Sater Haley and Ji reports also confirmed this view. Sater Haley 's research shows that blueberry has a unique blueberry flavor. ( 10.1016/J.SCIENTA.2021.110468). Ji et al.also believe that mature blueberry has a blueberry aroma, and the fruit has a sour taste after decay. (10.16377/j.cnki.issn1007-7731.2018.24.009)
Point 26: 4.0.This part is weak. Needs to be revise. Please provide all achievements separately now this big experiment is summarized too poorly.
Response 26: Revised. Thanks. Your opinion is very pertinent.

Round 2
Reviewer 1 Report
The authors have provided responses to many concerns raised; in general, the manuscript has been improved. At the same time, the explanations given revealed the need in additional clarifications.
1. Stage of coating application. The explanation added (l. 209-211) confirms that the coating was applied before harvest: "…after the coating solution had dried completely, blueberry fruit were picked". This detail is not obvious, and it is not clear from the manuscript in its present form. Moreover, the relevant section 2.6.1 is titled "Postharvest treatment" (l. 208). IF INDEED THE COATING WAS APPLIED PREHARVEST, this important detail should be stated clearly, maybe even reflected in the title: "… and preharvest edible coating", as well as in the abstract text: "…combined with preharvest edible coating… etc." The procedure of the coating application should be clarified in the M&M before the assays description. For example, the section 2.2 might be divided into two subsections: 2.2.1 Fabrication of TKL coating (not "film" because the film is formed on the fruit surface, after evaporation of the solvent, i.e. water), and 2.2.2 Coating application (instead of 2.6.1).
2. "Relative electrical conductivity". From the explanation provided (section 2.4 4) I understood that the technique described is the one known as "electrolyte leakage assay", widely used for characterizing membrane state, see for example Plant Physiology and Biochemistry, 2011, 49:1220-1227. This approach has been used also with blueberry fruit, e.g. Postharvest Biology and Technology, 88:88-95. For the better clarity, it may be recommended to use the above term and to interpret the results accordingly.
3. "The commodity rate" (l. 75). In their response to the reviewer, the authors explained this term, but it stays unclear to the readers. It may be recommended just to present the information in more general and universally understood terms, e.g. "improved the preservation", etc.
4. Language: the newly added sentences apparently have not been checked by a language editor and are sometimes rather obscure e.g. (just as an example) l. 434-436. Similarly, the "Total number of yeast and mold" should be presented as "Total count". Additional language check is needed.
Author Response
Point 1: Stage of coating application. The explanation added (l. 209-211) confirms that the coating was applied before harvest: "…after the coating solution had dried completely, blueberry fruit were picked". This detail is not obvious, and it is not clear from the manuscript in its present form. Moreover, the relevant section 2.6.1 is titled "Postharvest treatment" (l. 208). IF INDEED THE COATING WAS APPLIED PREHARVEST, this important detail should be stated clearly, maybe even reflected in the title: "… and preharvest edible coating", as well as in the abstract text: "…combined with preharvest edible coating… etc." The procedure of the coating application should be clarified in the M&M before the assays description. For example, the section 2.2 might be divided into two subsections: 2.2.1 Fabrication of TKL coating (not "film" because the film is formed on the fruit surface, after evaporation of the solvent, i.e. water), and 2.2.2 Coating application (instead of 2.6.1).
Response 1: Revised. Thanks, your opinion is very pertinent.
Point 2: "Relative electrical conductivity". From the explanation provided (section 2.4 4) I understood that the technique described is the one known as "electrolyte leakage assay", widely used for characterizing membrane state, see for example Plant Physiology and Biochemistry, 2011, 49:1220-1227. This approach has been used also with blueberry fruit, e.g. Postharvest Biology and Technology, 88:88-95. For the better clarity, it may be recommended to use the above term and to interpret the results accordingly.
Response 2: Revised. Thanks, your opinion is very pertinent.
Point 3: "The commodity rate" (l. 75). In their response to the reviewer, the authors explained this term, but it stays unclear to the readers. It may be recommended just to present the information in more general and universally understood terms, e.g. "improved the preservation", etc.
Response 3: Thanks, your opinion is very pertinent. Commodity rate is the ratio of the number of fruits available for commercial transactions to the total number of fruits. Fruits available for commercial transactions are unblemished fruit.
Point 4:. Language: the newly added sentences apparently have not been checked by a language editor and are sometimes rather obscure e.g. (just as an example) l. 434-436. Similarly, the "Total number of yeast and mold" should be presented as "Total count". Additional language check is needed.
Response 4: Revised. Thanks, your opinion is very pertinent.

Reviewer 3 Report
The authors have approached my comments professionally. Despite the large number of comments, they responded to each of them individually. Most have been taken into account by making appropriate additions or changes to the text. In my opinion, the quality of work has clearly increased and in its current form it is already suitable for publication in Foods.
Author Response
Thank you very much for your review